# Acceptance Model for Mobile Building Information Modeling (BIM)

**Sim-Hee Hong [1], Seul-Ki Lee [2], In-Han Kim [3] and Jung-Ho Yu [1],***

[1]    Department of Architecture Engineering, Kwangwoon University, Seoul 01897, Korea
[2]    ICEE, Seoul National University, Seoul 08826, Korea
[3]    Department of Architecture, Kyung-Hee University, Seoul 17104, Korea
*    Correspondence: myazure@kw.ac.kr; Tel.: +82-2940-5564

**Abstract:** Mobile Building Information Modeling (BIM) is noted for tools that enable the systematic interchange of information and contribute to enhancing collaborative performance through BIM. BIM programs, which are continuously available in the mobile environment, have been developed. Moreover, in some sites, mobile BIM is applied to generate benefits in projects. Various efforts are being made to use mobile BIM; however, its utilization is low. Also, mobile BIM has lacked an analysis of the factors that affect actual users' acceptance of mobile BIM. Therefore, this study analyzes the factors that affect the acceptance of mobile BIM by construction practitioners and presents the association of factors as a model to activate mobile BIM use. To this end, this study analyzed a literature review for suggesting the factors that were expected to affect mobile BIM acceptance. The assessment items were decided based on the analysis result. Second, 111 copies were received by surveying the construction practitioners. Third, it identified factors that significantly affected the acceptance of mobile BIM and proposed models through factor analysis and structural equation models. Finally, based on the analysis, it presented the findings. This study expects to contribute to enhanced acceptance of mobile BIM technology by managing the significant factors properly. Also, it is expected that the result can be used to develop a variety of mobile BIM that is more easily acceptable to them. This study presented a model for accepting mobile BIM based on the survey results of Korean practitioners; therefore, it is necessary to explore ways to generalize the model in the future.

**Keywords:** BIM; open BIM; mobile BIM; mobile application; technology acceptance model (TAM)

---

## 1. Introduction

### 1.1. Research Background

Since the 2000s, building information management (BIM) has been used in construction projects for various purposes, and the devices used for BIM have become more diverse. Initially, BIM was implemented on an office environment; however, BIM in the office environment is difficult to share, so real-time communication with the construction site was challenging. As a result, various technologies that enable real-time information use were applied, such as web-based cloud applications [1,2], virtual reality, and augmented reality [3,4]. Further, handheld devices, such as mobile phones, have enabled real-time information use [5].

The use of mobile devices in the construction field for information utilization is changing. Further, the way the project stakeholders operate is also changing. Mobile phones allow portability, thus enabling real-time communication between the system and the project stakeholders [6]. Mobile-based BIM can enhance the collaborative performance of BIM, thus enabling the linked exchange of information [7]. Mobile programs in construction are used for viewing and managing drawings and documents, safety

management, and communication. According to a survey by the Architectural Urban Research and Information Center (AURIC), approximately 40% of the respondents said they used mobile programs, and they used functions for managing drawings and documents, checking construction codes, checking drawings on the site, checking modifications in the construction situation, and so on [8].

With the development of various mobile programs for utilizing BIM, it is possible to quickly check and modify 3D information in real time. Since 2011, many companies have launched BIM programs that can be supported in mobile environments [8]. Most existing BIM tools were based on an office environment such as Revit and ArchiCAD; however, the use of mobile programs in the BIM field is increasing due to the development of tools based on mobile environments such as A360, BIMx, and Formit. Over time, it was observed that certain mobile BIM users reaped benefits in their projects, such as cost reduction and reduction of the rework rate [9]; therefore, there are efforts to improve mobile BIM usage even though the current utilization is low.

The reasons for this lack of mobile BIM usage in Korea were investigated, and it was noted that factors such as the age of the expected users [8] and inconsistencies between the actual working processes and BIM processes [10] were primarily responsible. However, another study reported that experience and age did not significantly affect the use of mobile BIM use [11]. Therefore, there is no overarching reason that can explain why mobile BIM as a technology has not been accepted by real users.

The technology acceptance model (TAM) defines external factors that affect the behaviors of users who accept a technology. Acceptance is defined as a psychological and technological state in which an individual or organization is prepared to use the technology smoothly. The TAM enables a definition of a significant factor that affects user accommodation of technology. TAMs are currently used in various fields, such as travel [12], sports [13], and shopping [14], among others, and are also used in the BIM field to analyze general BIM acceptance [15] and BIM acceptance in organizations [16,17]. If mobile BIM systems can also utilize the TAM, it is expected that an effective user-based technology acceptance analysis will be possible.

Therefore, this study aims to present significant findings on the activation of mobile BIM usage through a mobile BIM acceptance model through analyzing surveys of construction practitioners. First, we analyzed existing literature on mobile applications of the TAM and BIM TAM to find the external and internal variables that affect mobile BIM acceptance. Second, the hypothesis for the relationships between the variables was established based on the TAM2 and Information System (IS) models. Third, a survey was conducted based on the analysis results and hypothesis. Fourth, the results of the survey were analyzed using a structural equation, and the significant external variables for a mobile BIM TAM were suggested. Therefore, this study is expected to contribute toward basic research on the active adoption of mobile BIM technologies.

*1.2. Research Methodology*

As part of the efforts to improve the use of mobile BIM, this work analyzes the relationships among factors affecting acceptance of mobile BIM by construction practitioners and presents a mobile BIM TAM. The research processes followed are classified thusly.

(1) Define the mobile BIM characteristics based on a literature review: The characteristics of mobile applications and BIM were defined through a literature review to suggest TAMs. The analysis results were assessed for external factors that affected mobile BIM acceptance.

(2) Collect data to propose a mobile BIM TAM: The survey responses from construction practitioners (i.e., designers, contractors, Construction Managers (CMs), and BIM contractors) were collected. An overview of the data collection process is as follows.

The survey data were obtained from a sample of experienced BIM users or those experiencing BIM at work. The survey was conducted between 22 March and 29 April, and a total of 111 responses

were received. Each question was measured on a 7-point Likert scale ranging from "strongly disagree" to "strongly agree."

Among the 111 respondents, 31 were designers, 37 were from construction management organizations, 11 were from construction organizations, 18 were engineers, and 14 were researchers. The respondents' average experience in construction was about 11.6 years and the average experience with BIM was about 3.5 years. Their average preference for using BIM tools was 5.84 points on the 7-point Likert scale. The respondents' characteristics are detailed in Table 1.

(3) Explore key factors for mobile BIM acceptance through factor analysis: The survey results were analyzed as follows. A principal component analysis (PCA) of the factors was conducted using the software package IBM Statistics SPSS 21.0. The results are shown using the Kaiser–Meyer–Olkin (KMO) measure and Bartlett's test. Through this analysis, variables that did not have a significant impact on mobile BIM acceptance were removed, and the assessment items were classified as significant factors.

(4) Validate the proposed model using a structural equation model (SEM): A hypothesis was established for the relationships between the factors from the factor analysis results. An analysis of the SEM was conducted using the software IBM Statistics AMOS 21.0 to validate the hypothesis. The result shows the ratio of $\chi^2$ to the degrees of freedom (df), root-mean-square residual (RMR), parsimonious goodness of fit index (PGFI), Tucker–Lewis index (TLI), comparative fit index (CFI), and root-mean-square error of approximation (RMSEA). We eliminated the insignificant hypothesis of the set assumptions based on the path analysis results. Further, we checked the convergent validity and discriminant validity of the proposed model on the confirmatory factor analysis (CFA).

**Table 1.** Characteristics of the respondents (*n* = 111).

| Measure | | Frequency | % |
|---|---|---|---|
| the Respondent's Organization | Designer | 31 | 27.9 |
| | CM | 37 | 33.3 |
| | Contractor | 11 | 9.9 |
| | Engineer | 18 | 16.3 |
| | Researcher | 14 | 12.6 |
| Total | | | |
| Respondent's Average Experience | Construction Industry | Approx. 11.6 years | |
| | BIM | Approx. 3.5 years | |
| Preference for Using BIM Tools | | 5.84/7 | |

## 2. Literature Review

### 2.1. Mobile BIM in the Construction Industry

A mobile device is defined as a personal tool that is portable [18,19] and touchable [20,21]. Because mobile devices are portable, they enable checking and writing information regardless of the location. Devices referred to as being mobile include smartphones, smart pads, and service books [8], and mobile devices can use software applications that provide multipurpose convenience. Accordingly, mobile devices and applications allow users to add or delete functions according to their personal preferences while developing and using new applications as needed.

Mobile device use is increasing owing to characteristics such as the outdoor production of construction projects and the use of mobile devices by construction engineers to carry out field processes effectively [22]. Mobiles in construction are expected to contribute to reductions in construction time, capital costs, features, and components, while increasing the overall productivity [23].

BIM has been widely used in construction projects since the early 2000s. BIM is an integrated 3D object-based information storage system, which is used to pre-review tasks through model visualization,

such as building design, clash detection, and quantity calculation. Currently, the scope of the original BIM work has expanded considerably. BIM is also used in specialized fields such as structural analysis, energy analysis, and regulation checks. It is used in the design and construction phases, as well as in building inspection during the maintenance phase. However, it is difficult to utilize BIM in construction sites for real-time communication because the information is thus far generated and shared only in the office environment. Mobile BIM appears to improve the difficulty in sharing information and better utilizes BIM.

Mobile BIM is defined as a mobile-based system that can be used on smartphones, smart pads, and surface books, and provides an environment in which BIM data can be utilized regardless of the location [8]. For example, mobile BIM is available in the applications of A360, BIM 360, BIM 360 Ops, BIM 360 Field, BIMx, Field 3D, FINAL CAD, Formit, etc.

Mobile BIM can address many issues that arise due to the difficulty in sharing information between teams in the collaboration process. On some sites, mobile BIM has significantly reduced costs. For example, FINAL CAD, a Mobile BIM, reduces construction time by an average of 11%, rework rate by an average of 25%, and defect rate by an average of 50% at construction sites worldwide [9]. Mobile BIM can reduce the safety, paperwork, and drawing management by three times. Moreover, it is expected that a mobile environment will enable providing desired functions according to personal preferences based on the characteristics of the mobile devices themselves.

Various attempts have been made to use mobile BIM in practical applications. Mobile BIM is used on mobile platforms in certain projects for safety management or quality management by focusing on the construction phase; however, it is still underutilized. The reasons for low utilization of mobile BIM could possibly be the low usage of BIM tools and the age and experience of the users. However, other studies have noted that age and experience do not significantly affect the use of mobile BIM [24]. Low mobile BIM usage can also be interpreted as a lack of analysis regarding the factors that affect actual user acceptance and utilization.

*2.2. Theories Related Technology Acceptance*

To promote the use of mobile BIM, this study analyzed the factors and relationships affecting the acceptance of mobile BIM by construction practitioners based on the TAM and information system (IS) success model. This study considered the TAM, IS success model, model of innovation resistance (Ram, 1987), and diffusion theory (Rogers, 2003), among others, and adopted the TAM and IS success models to establish the research hypothesis.

The TAM is the most common theory to evaluate technology usage. The TAM expresses the relationships between factors that affect the acceptance of the technology. The IS success model is also the basis for other similar theories since it was revised in 2003. The IS success model expresses the factors that affect the use of technology. It is still used to evaluate technology such as a digital library [25], online learning [26], a mobile catering app [27], and so on.

Mobile BIM is used to improve an individual's work and enhance the organization's work, such as BIM. As a result, the variables associated with an organization must be reviewed to evaluate mobile BIM. Accordingly, we adopted TAM2 to evaluate social factors and the IS success model to evaluate the success of the information system. The other theories were excluded because the model of innovation resistance diffusion theory contained many variables that have similar meaning with the TAM and IS success models.

The TAM originates from the theory of reasoned action (TRA). It uses the constructs "perceived usefulness" and "perceived ease of use," which are the individual hypotheses that are affected by external variables. The TAM is used to indicate the relationships between external variables that affect users' acceptance of technology and the factors that affect their actual behaviors. This model assumes the relationships among the external variables, perceived usefulness, and perceived ease of use (see Figure 1). However, because the TAM is founded on the individual hypothesis, it has a limitation in that the effects of social influence are ignored.

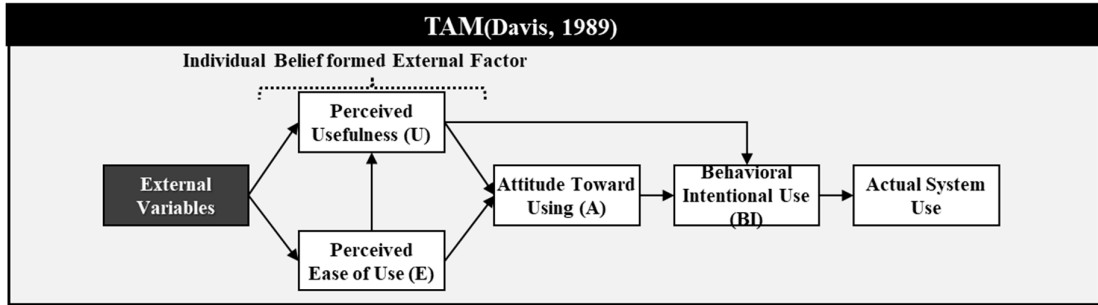

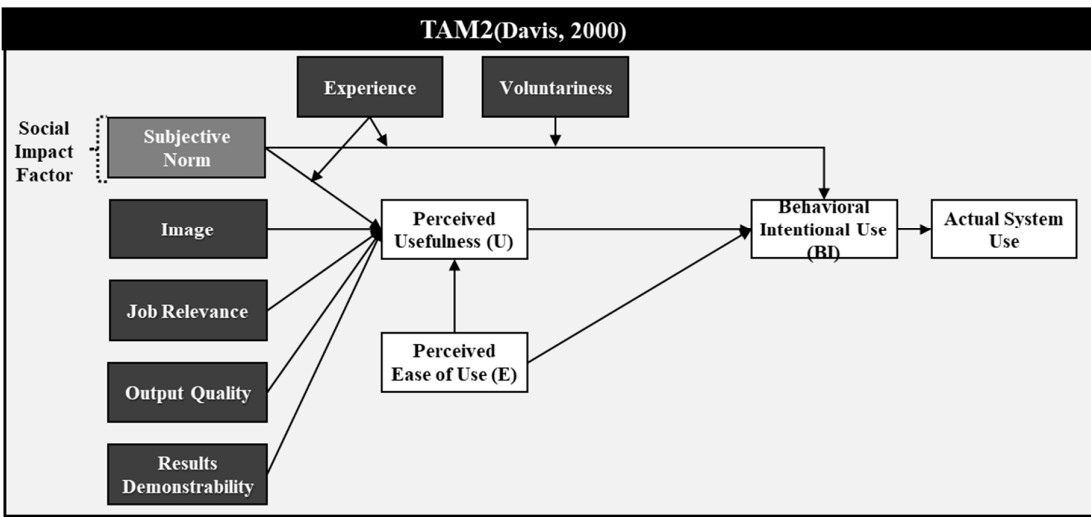

**Figure 1.** Davis' Technology Acceptance Model (TAM) (1998, 2000).

To improve this, Venkatesh and Davis [28] suggested the TAM2 (or ETAM) as an extension of the original TAM. TAM2 is included as external variables such as subjective norms, which are related to social influence [23]. The TAM2's external variables can be distinguished by both the social influence process and an identical tool process. The social influence process includes subjective norms, voluntariness, and image, whereas the identical tool process includes job relevance, output quality, and result demonstrability [29]. More recently, the TAM3 has been proposed, suggesting additional external variables that may affect perceived usefulness [30].

Also, DeLone and McLean proposed the IS success model [31,32] to assess the successful use of information systems. They proposed the original IS success model in 1992, which initially adopted system quality and information quality as the independent variables and expressed the impact as individual impact and organizational impact in four phases. Many researchers have expanded this model and the details; therefore, DeLone and McLean proposed an advanced IS success model in 2003 (see Figure 2).

In the advanced IS success model, service quality, which was perceived as a lower variable, was added as an independent variable, and the model was organized into three phases by integrating the individual and organizational impacts into net effects. The expanded IS success model is divided into six areas: system quality, information quality, service quality, use, user satisfaction, and net benefit.

Mobile BIM has the characteristics of BIM as an information system used for the benefit of an organization to enhance collaborative performance and to improve the sharing of information organically. Therefore, the research theory was established by referring to the expanded TAM2, including the social impact variables and the IS success model, which presented the success factors of the information management system. If essential factors that affect technology acceptance can be controlled, it is expected that the active adoption of such technology may be possible. Further, it is

expected that these results will assist mobile BIM developments that can more easily accommodate users by reflecting factors that affect their acceptance of technology.

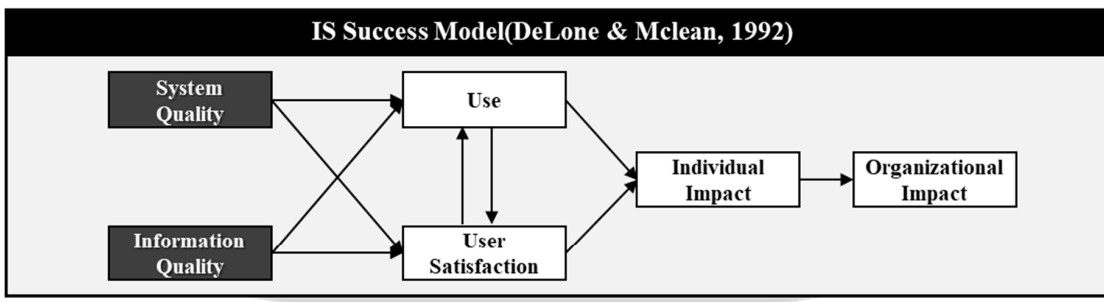

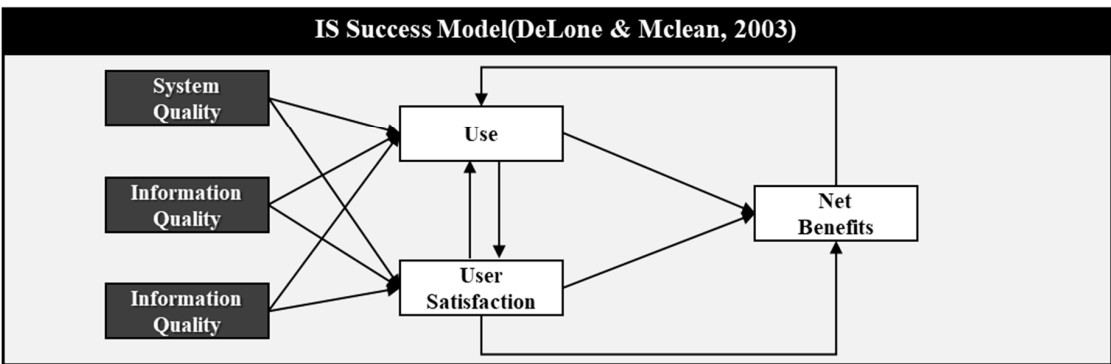

**Figure 2.** IS success model (1992, 2003).

## 3. Exploring the Key Factors for Mobile BIM Acceptance

### 3.1. Key Factors for Mobile BIM Acceptance

In this section, the exploratory factor analysis results are presented to establish a hypothesis for the mobile BIM acceptance model based on the survey results. The survey items comprise external variables that are expected to be significant to mobile BIM acceptance by referring to the TAM2 and IS success models, as well as previous studies related to the mobile TAM or BIM TAM. This study reviewed the relevant theories and literature reviews written in Korean and English related to the BIM TAM and mobile TAM that were published up until 2018 from around the world. Thus, the survey items are expected to be significant to construction practitioners worldwide. The representative literature is summarized in Table 2.

Literature reviews on the BIM TAM or mobile TAM aim to improve the users' skills and increase the amount of information that is targeted. First, mobile capability is based on constant internet access, ease of communication, and convenience of portability. The literature reviews related to mobile TAM adopted the quality and usability of application functions as external variables and verified the relationships between variables by adopting the perceived usefulness, usability, and use as the internal variables. Next, BIM improves individual convenience and organizational convenience using visual objects that contain information. The literature reviews related to the BIM TAM adopted external variables, not only on the quality of the system, but also on the organizational factors. Therefore, the external variables adopted the information technology characteristics in literature reviews.

Factors derived from literature reviews are summarized as follows: First, technology quality comprises key factors related to the performance of the tool that is being used, including system quality, information quality, service quality, and cost. Second, behavior control comprises key factors related to the environment in which the tool is expected to be used, including internal support, external support, internal pressure, and external pressure. Third, personal competency comprises key factors related

to the individual's psychological state toward the tool that is to be used, including self-efficiency and personal innovativeness. Fourth, organization competency comprises key factors related to the group psychological state toward the tool that is to be used, including collective efficiency and organization innovativeness.

**Table 2.** Literature Reviews on key factor for Mobile-BIM Acceptance.

| Category | Author | Details | Key Factors | Countries |
|----------|--------|---------|-------------|-----------|
| Mobile | [20] | Analyze the impact of mobile application characteristics on user satisfaction. | User usability, information quality, stability of security | Korea |
| Mobile | [18] | Presents factors that affect mobile application usage. Analyze the effects of factors using statistical analysis. | Utility, usability contents, entertainment, cost | USA |
| Mobile | [21] | Suggest factors that affect mobile application acceptance. Analyze the impact relationship of factors. | Usability, efficacy, innovativeness, security | Korea |
| Mobile | [11] | Analysis of key factors that affect the mobile application acceptance and present the findings. | Performance expectancy, effort expectancy, social influence, entertainment motivation, social utility motivation, communication motivation | Korea |
| Mobile | [33] | Present the acceptance model for a mobile service. | Trust, innovativeness, relationship drivers, functionality | USA |
| BIM | [34] | Analyze the major factors of BIM software selection to improve the use of BIM. | Usability, functionality, business, experience | USA |
| BIM | [15] | Analyze factors affecting BIM acceptance and the impact relationship of factors. | Technology quality, behavior control, personal competency, organizational competency, cost | Korea |
| BIM | [35] | Analyze the key factors in BIM acceptance from the point of view of CMs. Present the strategy of using BIM. | Social impact, personal impact, business impact | USA |
| BIM | [36] | Analyze critical success factors that affect the acceptance of BIM technology and present the findings. | Human-related factors, industry-related factors, project-related factors, policy-related factors, pesource-related factors | Korea |

### *3.2. Exploratory Factor Analysis of the Key Factors*

The survey conducted as part of this study collected 111 sets of responses from stakeholders in the construction industry (i.e., designers, contractors, CMs, engineers, and BIM contractors). A factor analysis was performed to remove external variables that did not significantly affect the acceptance of mobile BIM and its separate assessment items that had a significant impact on similar factors. First, a PCA was performed for factor extraction, and the varimax method with Kaiser normalization was performed as a factor rotation method. As a result, 10 items were eliminated from the 39 items that impeded the validity of the analysis. These excluded items are related to screening composition of system quality, items related to cost, and items related to the education of internal support systems. Second, the remaining 29 items were analyzed using factor analysis (see Table 3).

First, the KMO measure provides the statistical figure to which relationship between variables is well explained by the other variables, and it indicates the fitness of the variables selected by the factor analysis. In general, a KMO measure of 0.90 or higher is considered highly appropriate, and a KMO measure of less than 0.50 is considered unacceptable. The KMO measure of this study was 0.870. This implies that the 29 selected variables were appropriately selected. Next, Bartlett's test of sphericity checks whether a factor analysis model is appropriate by determining the model's goodness of fit with the *p*-value. Bartlett's test of sphericity showed a *p*-value less than 0.05; thus, the factor analysis model was assumed to be appropriate for the study. Meanwhile, the cumulative dispersion value of the data was 77.746%, and the results of factor analysis had a high description.

**Table 3.** Results of the factor analysis.

| Component | Items | Factor Loading | Eigenvalue | Cumulative % | Cronbach's $\alpha$ |
|---|---|---|---|---|---|
| Tool Quality | SVQ2 | 0.885 | 10.906 | 37.606 | 0.958 |
| | IQ3 | 0.868 | | | |
| | SVQ4 | 0.863 | | | |
| | SVQ3 | 0.854 | | | |
| | IQ2 | 0.844 | | | |
| | SVQ5 | 0.819 | | | |
| | IQ1 | 0.807 | | | |
| | IQ4 | 0.785 | | | |
| | SVQ1 | 0.772 | | | |
| | SYSQ2 | 0.708 | | | |
| Behavior Control | ES1 | 0.893 | 4.714 | 53.862 | 0.946 |
| | EP1 | 0.871 | | | |
| | EP2 | 0.861 | | | |
| | EP3 | 0.857 | | | |
| | IP1 | 0.808 | | | |
| | IS3 | 0.805 | | | |
| | IP2 | 0.796 | | | |
| | ES2 | 0.742 | | | |
| Personal Efficacy | PE2 | 0.922 | 3.218 | 64.957 | 0.897 |
| | PE4 | 0.887 | | | |
| | PE1 | 0.869 | | | |
| | PE3 | 0.839 | | | |
| | PI2 | 0.561 | | | |
| Organization Innovativeness | OI3 | 0.905 | 2.654 | 74.109 | 0.907 |
| | OI2 | 0.858 | | | |
| | OI1 | 0.806 | | | |
| Collective Efficacy | CE4 | 0.790 | 1.055 | 77.746 | 0.932 |
| | CE3 | 0.764 | | | |
| | CE1 | 0.696 | | | |
| Kaiser–Meyer–Olkin Measure of Sampling Adequacy | | | | | 0.870 |
| Bartlett's Test of Sphericity | | | Approx. Chi-Square | | 3172.681 |
| | | | df. | | 406 |
| | | | Sig. | | 0.000 |

SYSQ: System Quality, IQ: Information Quality, SVQ: Service Quality, IS: Internal Support, IP: Internal Pressure, ES: External Support, EP: External Pressure, PI: Personal Innovativeness, PE: Personal//; 07uiii efficiency, OI: Organization Innovativeness, CE: Collective efficiency

The first factor included 10 items, the second factor included 8 items, the third factor included 5 items, and the fourth and fifth factors included 3 items each. The first factor was named technology quality, the second was named behavior control, the third was named personal efficacy, the fourth was named organization innovativeness, and the fifth was named collective efficacy. These factors were used as the external variables in the TAM model. The details of each factor are described in Section 4.2: Research Hypotheses.

## 4. Proposed Mobile BIM Acceptance Model

### 4.1. Overview of the Proposed Model

The relationships among the key factors presented in the literature review and factor analysis were established in the form of a hypothesis to present a model for mobile BIM acceptance. First, the external variables adopted were tool quality, personal efficacy, behavior control, organization innovativeness, and collective efficacy, as derived from the results of the factor analysis. Second, the internal variables adopted were the ease of use and usefulness and consensus on appropriation

according to the IS success model and the characteristics of BIM. Finally, the intentions adopted were the individual intention and organization intention. Nine hypotheses on the relationships among the variables were established to suggest factors affecting mobile BIM acceptance (as shown in Table 4 and Figure 3).

**Table 4.** Research hypotheses.

| Hypotheses | | Definition |
|---|---|---|
| H1 | a | Tool quality will positively effect ease of use. |
| | b | Tool quality will positively effect usefulness. |
| H2 | a | Personal efficacy will positively effect ease of use. |
| | b | Personal efficacy will positively effect usefulness. |
| H3 | a | Behavior control will positively effect ease of use. |
| | b | Behavior control will positively effect usefulness. |
| H4 | a | Organization innovativeness will positively effect ease of use. |
| | b | Organization innovativeness will positively effect usefulness. |
| H5 | a | Collective efficacy will positively effect ease of use. |
| | b | Collective efficacy will positively effect usefulness. |
| H6 | a | Ease of use will positively effect usefulness. |
| | b | Usefulness will positively effect ease of use. |
| H7 | a | Ease of use will positively effect consensus on appropriation. |
| | b | Ease of use will positively effect individual intention. |
| | c | Ease of use will positively effect organizational intention. |
| H8 | a | Usefulness will positively effect consensus on appropriation. |
| | b | Usefulness will positively effect individual intention. |
| | c | Usefulness will positively effect organizational intention. |
| H9 | a | Consensus on appropriation will positively effect individual intention. |
| | b | Consensus on appropriation will positively effect organizational intention. |

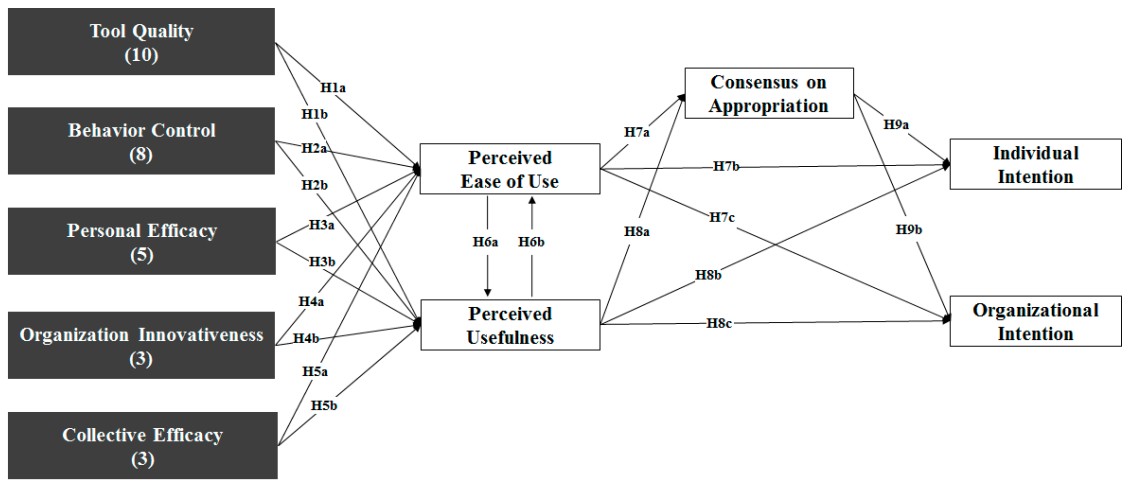

**Figure 3.** Proposed model for the mobile BIM acceptance model.

*4.2. Research Hypotheses*

4.2.1. External Variables for Mobile BIM Acceptance

The external variables presented in the literature review and factor analysis are as follows (shown in Table 5).

**Table 5.** Assessment items of external variables for Mobile-BIM acceptance.

| Variables | Assessment Items |
|---|---|
| **Tool Quality** | It is easy to input and output data with a mobile BIM. |
| | Using a mobile BIM improves the accessibility of information. |
| | The information in a mobile BIM is accurate and detailed. |
| | A mobile BIM provides sufficient information for the task. |
| | The information in a mobile BIM is available throughout the life cycle. |
| | A mobile BIM is easy to learn how to use. |
| | A mobile BIM is quick to respond to questions about how to use it. |
| | It is easy to use a mobile BIM manual. |
| | A mobile BIM is fast at reflecting user requirements. |
| | A mobile BIM enables continuous updating and After Service(A/S). |
| **Personal Efficacy** | I do not have any resistance toward using a mobile BIM. |
| | I can easily get used to using a mobile BIM. |
| | I understand the benefits of using a mobile BIM. |
| | I am confident that I will learn (manual, training, etc.) how to use a mobile BIM. |
| | I have the technical ability to use new information technology. |
| **Behavior Control** | Our organization offers incentives toward using a mobile BIM. |
| | Our organization enforces the use of a mobile BIM as a policy. |
| | Our boss or colleague requires the use of a mobile BIM. |
| | A mobile BIM enables economic benefits from the industry or the government. |
| | I can get appropriate training for the use of a mobile BIM from industry or government. |
| | Our organization is required to use a mobile BIM as a project delivery or contract method. |
| | Our organization requires the use of a mobile BIM in your relationship with your partner. |
| | Our organization is required to use a mobile BIM to meet the requirements of the contract. |
| **Organization Innovativeness** | Our organization has no psychological resistance to using new information technology. |
| | Our organization has technical capabilities for the use of new information technology. |
| | Our organization is active in the use of new information technology. |
| **Collective Efficacy** | Our organization does not have any resistance to the use of a mobile BIM. |
| | Our organization understands the benefits of using a mobile BIM. |
| | Our organization is confident in learning (mechanics, training, etc.) about how to use a mobile BIM. |

**Tool quality** is defined as the quality of the mobile BIM application. In this study, "technology" refers to the mobile BIM application, or the software running on mobile. Assessment items related to technology quality are system quality, information quality, and service quality.

System quality is an assessment item from the IS success model [30]. It has been presented as an external variable by many researchers [18,20,21,33,34,37]. The IS success model defines system quality as the quality of the measures of the information processing system itself [30], and the performance and convenience of the hardware and software [32]. Moreover, system quality has a similar meaning as job relevance in TAM2 and usability in the literature review.

Information quality is also an assessment item that is proposed in the IS success model [31]. It has been presented as an external variable by many researchers [15,35]. The IS success model defines information quality as measures of the information system output [31] and the quality of the

information provided by the information system [30]. Accordingly, information quality is defined as the quality of information related to work and the quality of the system environment in which the user obtains that information.

Service quality is a variable that is added as an independent variable by DeLone & McLean in 2003. Service quality is defined as the quality of the service that can be obtained from the work provided by the system and the relevant departments. Service quality is also presented as an external variable by various researchers [18,20].

**Personal efficacy** is defined as the psychological state of the individual who utilizes the mobile BIM application for their work. The assessment items are personal efficiency and personal innovativeness.

Personal efficiency is defined as a belief that an individual will be able to achieve successful results through action. Personal efficiency was adopted as an external variable to evaluate the use of technology [15,35].

Personal innovativeness is defined as an individual's active psychological attitude or ability to introduce new information technology. Personal innovativeness was adopted as an external variable to evaluate the use of technology [15,35].

**Behavior control** is defined as an external variable related to the environment in which the tool is used. Behavior control factors are divided into internal assessment items and external assessment items. The internal assessment items refer to those inside the organization, whereas the external assessment items refer to those outside the organization. The external variables related to the behavior control include pressure and support. First, pressure is defined as compulsory factors in the environment. It is similar to the subjective norm in TAM2. Subjective norm is the awareness of how people who are important to themselves will feel about doing certain things. It is adopted as an external variable in various literature reviews [21,28,33,35,36,38,39]. Next, support is defined as factors that help users to use their skills more efficiently.

Internal support is defined as the assistance within an organization from the organization's policies, bosses, or colleagues in selecting an action. For example, providing software in an organization, providing free education, and providing incentives.

Internal pressure is defined as a pressure that the organization gives to an individual, such as the organization's policies, bosses, or colleagues, in selecting an action. For example, compulsory policies in an organization, pressure from superiors or colleagues, and so on.

External support is defined as assistance from outside an organization through policies, systems, etc. in selecting an action. For example, there are government economic benefits, free government education, etc.

External pressure is defined as the pressure from outside the organization from the framework, contract, and relationship in selecting an action. For example, there is forced pressure from the owner, compulsory selection under the contract, and so on.

**Organization innovativeness** is defined as the organization's active psychological attitude or capacity to introduce new information technology. Organization innovativeness adopts an external variable to assess the use of technology in an organization as a factor from the characteristics of the BIM and IS success model [15,31,32].

**Collective efficacy** is defined as the belief that an organization will be able to achieve successful results through action. Collective efficacy also adopts an external variable to assess the use of technology in an organization from the characteristics of the BIM and IS success model [15,31,32].

4.2.2. Internal Variables for Mobile BIM Acceptance

In the mobile TAM, consensus on appropriation, perceived ease of use, and perceived usefulness were adopted as internal variables (shown in Table 6).

**Table 6.** Assessment items of internal variables for mobile BIM acceptance.

| Variables | Assessment Items |
|---|---|
| Consensus on Appropriation | The organization members show conformity on the tasks that apply a mobile BIM, which is set by the organization. |
| | The organization members show conformity regarding how to apply a mobile BIM, such as work guidelines and rules, which are set by the organization. |
| Perceived Ease of Use | It is easy to learn how to cooperate with a mobile-BIM. |
| | It is easy to exchange information with a mobile BIM. |
| | The guideline for collaboration with a mobile BIM is defined such that it can be followed quickly. |
| Perceived Usefulness | A mobile BIM improves to interoperability among stakeholders. |
| | A mobile-BIM allows for comprehensive management of life-cycle information. |
| | A mobile BIM reduces decision-making time. |
| | A mobile BIM can expand the utilization range of collaboration with other organizations. |
| | A mobile BIM reduces task-handling time. |
| | A mobile BIM improves task accuracy. |
| | A mobile BIM allows for easy collaboration with other organizations. |

**Consensus on appropriation** is defined as a consensus among members of the organization on information technology. Consensus on appropriation was chosen as an internal variable in the TAM for evaluating the BIM acceptance [15]. Mobile BIM was also selected as an internal variable because it also has characteristics similar to the BIM.

**Perceived ease of use** defines the extent to which it is believed that using an information technology system does not require much effort [40]. Perceived ease of use is presented as an internal variable in various literature reviews [18,21,33,35,39,41–43].

**Perceived usefulness** is defined as believing that the use of information technology systems will improve one's performance [39]. Perceived usefulness is presented as an internal variable in various literature reviews [18,21,33,35,39,41–43].

4.2.3. Intention toward Mobile BIM Acceptance

Intention is defined the extent of intended or planned use of an information technology system [40]. Intention is divided into individual intention and organizational intention according to the target. It is adopted as an internal variable in various literature reviews [18,21,33,35,39,41–43]. A variable similar to intention, net benefit, is employed by the IS success model [15,31,39,42] (shown in Table 7).

**Table 7.** Assessment items of individual and organizational intention to accept a mobile BIM.

| Variables | Assessment Items |
|---|---|
| Individual Intention | I have the intention to use a mobile BIM for my task. |
| | I have the intention to recommend a mobile BIM to others. |
| | I have the intention to take the time to learn how to use a mobile BIM. |
| Organizational Intention | My organization has the intention to encourage members toward using a mobile-BIM. |
| | My organization has the intention to be active in using a mobile BIM for the task. |
| | My organization has the intention to recommend a mobile BIM to other organizations that have a collaborative relationship with our organization. |

**Individual intention** is defined as an individual's intention or use plan to use an information technology system. If an internal variable has a significant effect on the individual intention, the individual's acceptance of the technology is also deemed significant.

**Organizational intention** is defined as an organization's intention or plan to use an information technology system. If the internal variables have a significant effect on the organizational intention, the organization's acceptance of technology is also deemed significant.

## 5. Model Validation

### 5.1. Validation of the Proposed Model

In this chapter, the proposed model is validated as a structural equation model under key factors, which were presented as an exploratory factor analysis. To validate our measurement model, we analyzed the assessments of convergent and discriminant validity. This study used the following model fit measures to decide the model's goodness-of-fit: the ratio of $\chi^2$ to df, RMR, PGFI, the TLI, CFI, and RMSEA. The model-fit indices of the proposed model and the acceptance level were compared.

The initial measurement model did not show a good fit compared with the recommended values of $\chi^2/df = 2.773$, RMR = 0.312, PGFI = 0.365, TLI = 0.617, CFI = 0.635, and RMSEA = 0.131. Therefore, organization innovativeness and collective efficacy were removed. This was because the relationship between the latent variables in the initial measurement model was not significant, and the standardized factorial load was less than 0.5. In addition, we re-analyzed behavior control, which was smaller than the *p*-value and was excluded from some hypotheses, by dividing it into internal support, internal pressure, external support, and external pressure. The analysis results show that internal support and internal pressure were significant to the model. Further, consensus on appropriation and organizational intention were also excluded as factors that impeded the model's suitability. Therefore, the final measurement model had the values of $\chi^2/df = 2.08$, RMR = 0.45, PGFI = 0.553, TLI = 0.854, CFI = 0.867, and RMSEA = 0.102 (shown in Table 8).

**Table 8.** Fit indices for research model.

| Fit Indices | Recommended Value | Measurement Model | Structural Model |
|:---:|:---:|:---:|:---:|
| $\chi^2/df$ | ≤3.0 | 2.08 | 2.02 |
| RMR | ≤0.1 | 0.145 | 0.145 |
| PGFI | ≥0.5 | 0.553 | 0.504 |
| TLI | ≥0.9 | 0.854 | 0.73 |
| CFI | ≥0.9 | 0.867 | 0.75 |
| RMSEA | ≤0.1 | 0.102 | 0.095 |

The SEM analysis showed that tool quality, personal efficacy, and internal control had a positive effect on the ease of use and usefulness. In other words, if the tool quality, personal efficacy, and internal control increased, the ease of use and usefulness also increased. Further, usefulness had a significant positive effect on individual intention; the individual intention increased with usefulness. The CFA results showed that the path from tool quality, personal efficacy, and behavior control among internal support to individual intention was significant (shown in Table 9).

The discriminant validity test was not satisfied between the factors of ease of use and technology quality (Table 10). However, technology quality was not removed because it has been referred to as a significant factor in literature reviews.

**Table 9.** Results of CFA.

| Latent Constructs | Observed Indicators | Factor Loading | *t*-Value | Composite Reliability | AVE |
|---|---|---|---|---|---|
| Tool Quality (TQ) | TQ 1 | 0.837 | - | 0.906 | 0.493 |
| | TQ 2 | 0.883 | 11.795 | | |
| | TQ 3 | 0.897 | 12.117 | | |
| | TQ 4 | 0.905 | 12.326 | | |
| | TQ 5 | 0.813 | 10.253 | | |
| | TQ 6 | 0.788 | 9.767 | | |
| | TQ 7 | 0.863 | 11.315 | | |
| | TQ 8 | 0.833 | 10.664 | | |
| | TQ 9 | 0.786 | 9.726 | | |
| Personal Efficacy (PE) | PE 1 | 0.723 | 8.593 | 0.865 | 0.571 |
| | PE 2 | 0.472 | - | | |
| | PE 3 | 0.855 | 5.014 | | |
| | PE 4 | 0.83 | 4.962 | | |
| | PE 5 | 0.936 | 5.16 | | |
| | PE 6 | 0.904 | 5.108 | | |
| Behavior Control (BC) | BC 1 | 0.949 | - | 0.856 | 0.667 |
| | BC 2 | 0.925 | 14.711 | | |
| | BC 3 | 0.750 | 9.994 | | |
| Perceived Ease of Use (EOU) | EOU 1 | 0.898 | - | 0.675 | 0.409 |
| | EOU 2 | 0.887 | 13.068 | | |
| | EOU 3 | 0.885 | 13.004 | | |
| Perceived Usefulness (U) | U 1 | 0.919 | - | 0.904 | 0.535 |
| | U 2 | 0.938 | 17.488 | | |
| | U 3 | 0.888 | 14.768 | | |
| | U 4 | 0.882 | 14.527 | | |
| | U 5 | 0.899 | 15.297 | | |
| | U 6 | 0.722 | 9.495 | | |
| | U 7 | 0.895 | 15.113 | | |
| Individual Intention to Accept Mobile-BIM (IIA) | IIA 1 | 0.971 | - | 0.800 | 0.572 |
| | IIA 2 | 0.848 | 13.025 | | |
| | IIA 3 | 0.805 | 11.648 | | |

\* *t*-Value for these parameters were not available because they were fixed for scaling purposes.

**Table 10.** Results of discriminant validity test.

| Observed Indicators | | r2 | | AVE | Discriminant Validity |
|---|---|---|---|---|---|
| Tool Quality (TQ) | PE | 0.094 | | 0.571 | Acceptable |
| | BC | 0.089 | | 0.667 | Acceptable |
| | EOU | 0.590 | 0.493 | 0.409 | Unacceptable |
| | U | 0.543 | | 0.535 | Unacceptable |
| | IIA | 0.213 | | 0.572 | Acceptable |
| Personal Efficacy (PE) | BC | 0.001 | | 0.667 | Acceptable |
| | EOU | 0.200 | 0.571 | 0.409 | Acceptable |
| | U | 0.138 | | 0.535 | Acceptable |
| | IIA | 0.348 | | 0.572 | Acceptable |
| Behavior Control (BC) | EOU | 0.0847 | | 0.409 | Acceptable |
| | U | 0.117 | 0.667 | 0.535 | Acceptable |
| | IIA | 0.024 | | 0.572 | Acceptable |
| Perceived Ease of Use (EOU) | U | 0.605 | 0.409 | 0.535 | Unacceptable |
| | IIA | 0.272 | | 0.572 | Acceptable |
| Perceived Usefulness (U) | IIA | 0.430 | 0.535 | 0.572 | Acceptable |

As expected, hypothesis H1a and H2a were supported ($\gamma = 0.666$ and $\gamma = 0.42$, respectively). Increased tool quality and personal efficacy were associated with increased perceived ease of use. Behavior control was not very supported ($\gamma = 0.123$). The tool quality and personal efficacy had an effect on perceived usefulness. H1b and H2b were supported ($\gamma = 0.424$ and $\gamma = 0.45$, respectively). Perceived ease of use appeared to be a significant determinant of perceived usefulness; H6a was supported ($\beta = 0.405$). In addition, perceived usefulness had a significant effect on individual intention in accepting a mobile BIM; H8b was supported ($\beta = 0.631$). Therefore, tool quality and personal efficacy exhibited a relatively strong influence on mobile BIM acceptance, and behavior control had relatively weak influence. H1a, H2a, and H3a explained 58.6% of the variance in the perceived ease of use. H1b, H2b, and H6a explained 71.6% of the variance in perceived usefulness. H8b explained 43.7% of the variance in individual intention to accept a mobile BIM. The final proposed model is shown in Figure 4, and the direct, indirect, and total effects of each construct are summarized in Table 11.

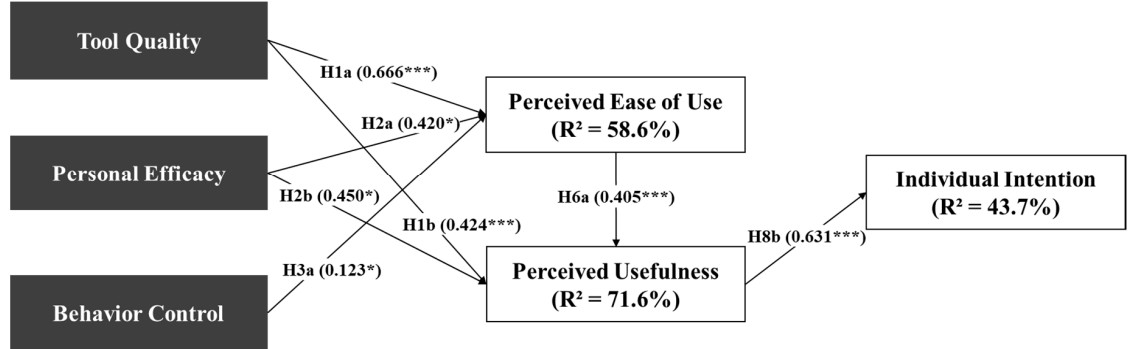

**Figure 4.** The validation results of the proposed model. $^{+}$ $p < 0.1$, $^{*}$ $p < 0.05$, $^{**}$ $p < 0.01$, $^{***}$ $p < 0.001$.

**Table 11.** Direct, indirect and total effects.

|  | Total Effects | | | Direct Effects | | | Indirect Effects | | |
|---|---|---|---|---|---|---|---|---|---|
|  | EOU | U | IIA | EOU | U | IIA | EOU | U | IIA |
| TQ | 0.666 | 0.693 | 0.449 | 0.666 | 0.424 | 0 | 0 | 0.269 | 0.449 |
| PE | 0.42 | 0.619 | 0.398 | 0.42 | 0.45 | 0 | 0 | 0.17 | 0.398 |
| BC | 0.123 | 0.071 | 0.047 | 0.123 | 0.021 | 0 | 0 | 0.05 | 0.047 |
| EOU | 0 | 0.405 | 0.272 | 0 | 0.405 | 0.017 | 0 | 0 | 0.255 |
| U | 0 | 0 | 0.631 | 0 | 0 | 0.631 | 0 | 0 | 0 |
| IIA | 0 | 0 | 0 | 0 | 0 | 0 | 0 | 0 | 0 |

## 5.2. Findings and Discussion

As a result of the analysis, the external factors that significantly affected the mobile BIM acceptance were found to be technology quality, self-efficiency, and behavior control.

**Tool Quality:** Tool quality is the software performance of the mobile BIM application, suggesting that the quality of the tools that the users need for the mobile BIM applications must be ensured. As a result, it is expected that the application could contribute to the mobile BIM acceptance if the application reflects the assessment items of the technology quality analyzed in this study as being significant to the mobile BIM acceptance. First, the mobile BIM should facilitate the input and output of data and provide reasonable access to information. This is consistent with the mobile nature of having easy access to information regardless of the location. Moreover, a variety of ways to efficiently use the information on mobile hardware with small screens, such as a user-centered interface design, should be explored. Second, the information that can be obtained from the tool should be accurate and detailed. The information obtained should be available throughout the life cycle. This conforms to the characteristics of the BIM that detailed information generated during the project life cycle can be aggregated based on visual objects. Accordingly, various methods should be explored to effectively

manage various information, such as BIM information, construction project information, and web cloud utilization. Third, it should be easy to understand the instructions and learn how to use. Replies to questions about how to use them should be answered quickly. Accordingly, there is a need to provide manuals or tutorials to help the users learn how to use the system. Moreover, there should be a window for exchanging questions and answers on how to use it. Finally, continuous updating of tools and A/S should be possible because users require speed. This suggests the need for a steady system performance improvement and suggests that a community should be provided for continuous feedback from the implementer on the system performance.

**Personal Efficacy:** Personal efficiency is the psychological state of an individual using the mobile BIM application in his or her work. It suggests that the mobile BIM induces a belief that the use of a mobile BIM will produce successful results. As a result, measures should be taken to appropriately promote various benefits that individuals can enjoy by using the mobile BIM.

**Behavior Control:** Behavior control is the environment in which tools are used. It suggests that mobile BIM applications require internal support and internal pressure, which are appropriate to the internal structure of the organization. First, the mobile BIM can be more easily used when receiving ancillary help from an organization's policies, members, etc. This suggests that appropriate policies should be established in the organization for the use of a mobile BIM, such as free educational opportunities and free software. Second, a mobile BIM can be more easily used when it is under appropriate pressure from the organization's policies, members, etc. This suggests that organizations, such as providing incentives, should have appropriate policies to encourage individuals to use tools, as well as provide assistance in using the mobile BIM.

Further, unlike the BIM TAM [15–17], it has been found that the factors associated with the organization do not affect the acceptance of the technology. A mobile BIM is a tool that organizations already use as part of their efforts to make BIM more comfortable to use. Accordingly, the acceptance of the mobile BIM technologies focuses on enhancing the ease of use for individuals, unlike the decision-making that determines the use of the BIM tools in groups without BIM. Thus, unlike BIM, areas related to the organization are interpreted as not affecting technology acceptance.

## 6. Conclusions

Mobile devices in construction help to operate an information system regardless of the location, and it is introduces many changes to information approach and to the working of project stakeholders. Besides, mobile BIM draws attention to tools that can contribute to enhancing collaborative performance through BIM by enabling the exchange of information organically. Accordingly, BIM programs supported by the mobile environment are continuously being released, and it is found that the mobile BIM applied to some sites generates actual benefits in carrying out projects.

Although various efforts are being made to use a mobile BIM, it is found that it is underutilized due to the lack of experience in using a mobile BIM, the high age of expected users, and inconsistency between the actual working processes and BIM processes. Moreover, some studies related to the use of mobile BIM have differing views on the same cause, which may be interpreted as a lack of analysis of factors that affect the actual users' acceptance and utilization of the mobile BIM.

Accordingly, this study analyzed the factors and their associations with the factors that affect the acceptance of a mobile BIM by construction practitioners as part of its efforts to activate the use of the mobile BIM. A questionnaire was organized to define the characteristics of the mobile and BIM through existing literature considerations and to analyze the external factors affecting the mobile BIM acceptance. The survey collected 111 copies from construction practitioners (designers, contractors, CMs, engineering, and BIM contractors). The results of the survey were broken down into factors and path analysis was undertaken using SPSS 21.0 and Amos 21.0, and the factors that significantly affected mobile BIM acceptance were expressed in models through the structured model by checking the concentration and determination feasibility.

As a result of the analysis, external factors that significantly affected the mobile BIM acceptance were found to be tool quality, personal efficacy, and behavior control. First, tool quality is the software performance of a mobile BIM application, suggesting that users need the quality of the tools they need for mobile BIM. Second, personal efficacy suggests that measures should be taken to appropriately promote the various benefits of using a mobile BIM. Third, behavior control suggests that there must be adequate support and pressure within the organization to use a mobile BIM.

It is expected that this study will contribute to the revitalization of the technology along with the acceptance of the technology, provided that significant factors affecting the acceptance of the technology are appropriately managed. Moreover, it is expected that a mobile BIM will be developed to allow users to develop a variety of applications that are more easily acceptable to them.

This study is limited in that it was difficult to generalize these results for all countries because our survey subject of this study is construction practitioners in Korea. If future studies conduct surveys on USA or Europe where the use level of mobile BIM is high, it will be able to present new findings due to comparing the results.

**Author Contributions:** Conceptualization, S.-H.H., S.-K.L., and J.-H.Y.; methodology, S.-H.H., S.-K.L., and J.-H.Y.; software, S.-K.L.; validation, S.-H.H., S.-K.L., I.-H.K. and J.-H.Y.; formal analysis, S.-H.H. and S.-K.L.; investigation, S.-H.H. and S.-K.L.; resources, S.-H.H., I.-H.K. and J.-H.Y.; data curation, S.-H.H. and S.-K.L.; writing—original draft preparation, S.-H.H.; writing—review and editing, S.-K.L., and J.-H.Y.; visualization, S.-H.H. and S.-K.L.; supervision, J.-H.Y.; project administration, J.-H.Y.; funding acquisition, I.-H.K. and J.-H.Y.

**Funding:** This research was supported by a grant (19AUDP-B127891-03) from the Architecture & Urban Development Research Program funded by the Ministry of Land, Infrastructure and Transport of the Korean government.

**Conflicts of Interest:** The authors declare no conflict of interest.

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
