# Peer review of "Acceptance Model for Mobile Building Information Modeling (BIM)"

_applsci, doi:10.3390/app9183668_

Round 1
Reviewer 1 Report
In general, it's a qualified paper with fine structures, methodologies, analyses and detailed simulation on the proposed methodology. Authors propose a mobile MBI model to deal with the factors that affect the acceptance of mobile BIM by construction practitioners. Also, this study defines the mobile BIM characteristics based on literature review, collects data to propose mobile BIM TAM, explores key factor for mobile BIM acceptance using factor analysis, and validates the proposed model using structural equation model. Eventually, authors analyze the technology quality, self-efficiency, and behavior control and show that significant factors affecting the acceptance of technology are appropriately managed. I therefore suggest this paper can be accepted. I also recommend authors can compare more related methods with this proposed scheme.
Author Response
▪ Comment 1:
In general, it's a qualified paper with fine structures, methodologies, analyses and detailed simulation on the proposed methodology. Authors propose a mobile MBI model to deal with the factors that affect the acceptance of mobile BIM by construction practitioners. Also, this study defines the mobile BIM characteristics based on literature review, collects data to propose mobile BIM TAM, explores key factor for mobile BIM acceptance using factor analysis, and validates the proposed model using structural equation model. Eventually, authors analyze the technology quality, self-efficiency, and behavior control and show that significant factors affecting the acceptance of technology are appropriately managed. I, therefore, suggest this paper can be accepted. I also recommend authors can compare more related methods with this proposed scheme.
⇒ Response to comment 1
We agree with your comment and recommendation.
Our survey subject of this study is construction practitioners in Korea. There is a limitation that it is difficult to generalize and apply the findings directly.
If we conduct a survey on the U.S. or Europe where the use level of Mobile BIM is high, future study will be able to present new findings due to comparing the results.
We added the manuscript in section 6, line 483-486.

Reviewer 2 Report
The article takes a very actual and interesting topic. The main goal is questionnaire surveys, but they are the original contributions of the authors.
However, some elements require improvement. It is necessary to extend the literature review and introduction to research and articles from around the world, and not just from the "Asian" cultural circle. In particular, the research and use of mobile BIM in the US and Europe is already at a high level.
Table 2 requires a description of the countries in which the described applications for Mobile-BIM are used and supplemented with world literature.
Table 3 also needs to be explained. A broader description of the test results contained in Table 3 is necessary, and at least the Kaiser-Meyer-Olkin (KMO) and Bartlett's test of sphericity brief description.
I consider the article to be interesting and suitable for publication after minor adjustments.
Author Response
▪ Comment 1:
It is necessary to extend the literature review and introduction to research and articles from around the world, and not just from the "Asian" cultural circle. In particular, the research and use of mobile BIM in the US and Europe are already at a high level.
⇒ Response to comment 1
This study reviewed the relevant theories and literature reviews written in Korean and English related to BIM TAM and Mobile TAM that were published until 2018 from around the world. Therefore, our survey Item is not limited to Asian Cultural Circle.
Our survey subject of this study is construction practitioners in Korea. There is a limitation that it is difficult to generalize and apply the findings directly.
If we conduct a survey on the U.S. or Europe where the use level of Mobile BIM is high, future study will be able to present new findings due to comparing the results.
We added the manuscript in section 6, line 483-486.
▪ Comment 2:
Table 2 requires a description of the countries in which the described applications for Mobile-BIM are used and supplemented with world literature.
⇒ Response to comment 2
In order to make the survey item, we reviewed the domestic BIM TAM and Mobile TAM literature reviews until 2018 and the worldwide BIM and Mobile TAM literature reviews written in English until 2018.
We analyzed about twenty literature reviews, and ten representative literature reviews were suggested in Table 2.
We have added a description of the countries in Table 2.
▪ Comment 3:
Table 3 also needs to be explained. A broader description of the test results contained in Table 3 is necessary, and at least the Kaiser-Meyer-Olkin (KMO) and Bartlett's test of sphericity brief description.
⇒ Response to comment 3
We have amended Kaiser-Meyer-Olkin (KMO) and Bartlett's test of sphericity brief description related to Table 3. Please refer to section 3.2, line 247-255 of the resubmitted manuscript.

Reviewer 3 Report
There are major shortcomings that need fixing, most importantly:
The research problem is not clearly defined and the rationale/significance of the article is not explained (i.e. why investigating about acceptance while the paper also states that the technologies for mobile BIM are not market-ready; one can argue that acceptance will come naturally when necessity and readiness are sufficient).
Most literature references are older than 4 years and fail to reflect the latest developments in BIM and mobile technologies. The literature on information system success by DeLone et al. is dated from 1992, long before the era of BIM and mobile technologies. The poor selection of literature references renders the lack of credibility of the literature review.
Mobile BIM is described in a too general way (i.e. no in-depth explanation about the features and techniques which are available and in development which determine the market/user acceptance).
It is doubtful whether the term of PC (personal computer) is used appropriately to describe the contrast between mobile and desktop systems. It seems that the article only consider mobile phones as mobile devices (see line 39). What about tablets (like Microsoft Surface) and 2-in-1 laptops?
The writing style is poor as different topics are presented without a clear storyline (for example: the alinea starting at line 53 had no connection to the preceding alinea; this poor structure appears throughout the article).
Hypothesis (section 4.2) and Findings (section 5.2) are invalid as they just present the definitions / descriptions of the factors.
The conclusions do not present a new insight.
It is doubtful whether the selected analytical method and model validation for the survey fit the composition and the number of samples. It seems that the authors give too much attention to experiment with the complicated analytical method/model. The topic of mobile BIM seems to be of a secondary interest just for the sake of testing/applying the analytical method/model.
It is recommended to add a section in the beginning of the paper with brief definitions of the technical terms and acronyms used in the article. Currently, the definitions of the terms are scattered throughout the article, even in section 4.2 and 5.2.
Some sentences are not expressed in a proper English writing style. Language editing is needed.
Author Response
▪ Comment 1:
The research problem is not clearly defined and the rationale/significance of the article is not explained (i.e. why investigating about acceptance while the paper also states that the technologies for mobile BIM are not market-ready; one can argue that acceptance will come naturally when necessity and readiness are sufficient).
⇒ Response to comment 1
Mobile BIM has characteristics to be able to communicate using BIM information regardless of location. If mobile BIM used, it enables actual benefits such as cost reduction, time reduction, and so on. However, Korea is underutilized as it is not ready to accept Mobile BIM.
Acceptance is defined as a psychological and technological state in which an individual or organization is prepared to use the technology smoothly. The reason why this meaning of acceptance is difficult is the limitations of the user, rather than technical problems.
In order to find out factors affecting acceptance and what mechanisms the factors influence acceptance, this study conducted three things for accepting mobile BIM.
First, this study suggested factors affecting acceptance that must be prepared to accept Mobile BIM in the Korean construction industry. Second, we defined and validated the mobile BIM acceptance model based on the TAM, the most basic research model, which defined the process that users accepted a particular technology. Third, based on a proven model, we suggested the findings on what to be ready for using mobile BIM smoothly from the perspective of individuals and organizations.
We have amended to define the research problem more clearly in section 1.1 line 70-71.
▪ Comment 2:
Most literature references are older than 4 years and fail to reflect the latest developments in BIM and mobile technologies. The literature on information system success by DeLone et al. is dated from 1992, long before the era of BIM and mobile technologies. The poor selection of literature references renders the lack of credibility of the literature review.
⇒ Response to comment 2
This study considered TAM, IS success model, Model of Innovation Resistance, Diffusion Theory, among others, and adopted TAM and IS access model to establish the research hypothesis.
TAM is the most common theory to evaluate technology usage and IS success model is also the basis for other similar theories since it revised in 2003. It is still used to evaluate the technology such as Digital library (Ahmed I. A. et al., 2017), Online clearing (Osama Isaac et al., 2019), Mobile handling app (Yi-Shun Wang et al. 2019) and so on.
Mobile BIM is used not only to improve the individual's work but also to enhance the organization's work like BIM. As a result, it needs to review the variables associated with an organization to evaluate mobile BIM. Accordingly, we adopted TAM2 to evaluate social factors and IS success model to evaluate the success of the information system.
We added recent literature review related Mobile TAM and BIM TAM in Section 2.2 line 155-170.
▪ Comment 3:
Mobile BIM is described in a too general way (i.e. no in-depth explanation about the features and techniques which are available and in development which determine the market/user acceptance).
It is doubtful whether the term PC (personal computer) is used appropriately to describe the contrast between mobile and desktop systems. It seems that the article only considers mobile phones as mobile devices (see line 39). What about tablets (like Microsoft Surface) and 2-in-1 laptops?
⇒ Response to comment 3
We have added the manuscript related to the purpose of mobile BIM, the status of mobile BIM development, and the economic effects of mobile BIM use in section 1.1 line 38-53 and 2.1 line 140-146.
Mobile BIM also includes a smartphone, smart pad (tablets), surface book, etc. So, it also included in Mobile such as tables and surface book that used personally.
In the first version of the manuscript, the term PC was used to refer to not portable computer environment in the office environment. However, Mobile can also be described as a PC; the term PC has been modified.
The definition is more clearly described in section 2.1 line 136-139
▪ Comment 4:
Hypothesis (section 4.2) and Findings (section 5.2) are invalid as they just present the definitions/descriptions of the factors. The conclusions do not present a new insight.
It is doubtful whether the selected analytical method and model validation for the survey fit the composition and the number of samples. It seems that the authors give too much attention to experiment with the complicated analytical method/model. The topic of mobile BIM seems to be of a secondary interest just for the sake of testing/applying the analytical method/model.
⇒ Response to comment 4
In order to find out factors affecting acceptance and what mechanisms the factors influence acceptance, we validated the mobile BIM acceptance model based on the TAM, the most basic research model, which defined the process that users accepted a particular technology.
Based on the proven model, we suggested findings on what to be ready for using mobile BIM smoothly from the perspective of individuals and organizations.
We have added the details related to set up the hypothesis in section 4.1. Also, the findings are re-explained based on the analysis results in section 5.2.
▪ Comment 5:
It is recommended to add a section at the beginning of the paper with brief definitions of the technical terms and acronyms used in the article. Currently, the definitions of the terms are scattered throughout the article, even in section 4.2 and 5.2.
⇒ Response to comment 5
We have added that the abbreviations of statistical terms are included in the research methodology in section 1.2., line 101-113.
▪ Comment 6:
Some sentences are not expressed in a proper English writing style. Language editing is needed
⇒ Response to comment 6
Grammar and expression errors are corrected through professional proofreading and editing.

Round 2
Reviewer 3 Report
The article has been sufficiently improved. Thank you to the authors for taking the review comments into account.